# Echinacoside as a Novel Ferroptosis Inducer in Hepatocellular Carcinoma: Mechanistic Insights from TP53/SLC7A11/GPX4 Pathway Modulation

**DOI:** 10.3390/ijms27010411

**Published:** 2025-12-30

**Authors:** Pei Wang, Jianhao Lin, Deqi Su

**Affiliations:** Institute of Medical Sciences, School of Public Health, Xinjiang Medical University, Urumqi 830017, China; peiwang26@163.com (P.W.); ljh12184223@163.com (J.L.)

**Keywords:** echinacoside, hepatocellular carcinoma, ferroptosis, network pharmacology

## Abstract

Despite the known antitumor properties of echinacoside (ECH), its specific role and mechanism in hepatocellular carcinoma (HCC) require in-depth exploration. Our study aimed to decipher the mechanism of ECH against HCC through a multi-disciplinary strategy. We first identified tumor protein p53 (TP53) as a key mediator and ferroptosis as a critical process, through network pharmacology and enrichment analyses. The direct interaction between ECH and TP53 was validated by molecular docking and dynamics simulations. In vitro assessments demonstrated that ECH suppresses HCC proliferation by activating ferroptosis, marked by increased intracellular Fe^2+^, lipid peroxidation (LPO), and malondialdehyde (MDA), alongside reduced glutathione (GSH). The ferroptosis inhibitor ferrostatin-1 notably attenuated ECH’s effects, confirming ferroptosis as the primary mode of cell death. Further mechanistic investigation revealed that ECH acts through the TP53/solute carrier family 7 member 11(SLC7A11)/glutathione peroxidase 4(GPX4) pathway. These results collectively identify ECH as a promising ferroptosis-inducing agent for HCC therapy via TP53 activation.

## 1. Introduction

Primary liver cancer is a common malignancy that poses a serious threat to human health. Hepatocellular carcinoma (HCC) accounts for 80–90% of primary liver cancer cases [1]. The distribution of HCC exhibits significant geographical variation, with the highest incidence rates observed in Asia, particularly in East and Southeast Asian countries. China alone accounts for nearly half of the global cases [2,3].

The primary treatment options currently available for HCC include hepatectomy, ablation therapy, chemotherapy, radiotherapy, liver transplantation, immunotherapy, and molecularly targeted drugs [4]. Nonetheless, these treatment regimens are associated with adverse reactions, high recurrence and metastasis rates, and significant off-target effects [5,6]. While some targeted drugs, such as sorafenib, demonstrate relatively good efficacy, most options are prohibitively expensive, and most patients develop resistance within six months of treatment [7]. This indicates that the current clinical management of HCC is suboptimal and highlight the need to elucidate HCC mechanisms and to develop innovative therapeutic strategies.

Ferroptosis, a form of iron-dependent regulated cell death, was first formally described by Dixon et al. in 2012 [8], and differs mechanistically and morphologically from conventional cell death pathways, such as apoptosis and necrosis. Ferroptosis primarily causes cell death by disrupting the structure and function of the cell membrane through the accumulation of phospholipid peroxides [9,10]. Moreover, compared to normal physiological hepatocytes, HCC cells exhibit high expression of the RAS oncogene and increased iron content, indicating heightened sensitivity to ferroptosis [11]. Combining ferroptosis inducers with immune checkpoint inhibitors significantly enhances therapeutic efficacy against HCC [12]. Research findings indicate that the iron chelator deferoxamine depletes intracellular iron stores, providing significant protection to HCC cells against the cytotoxic effects of sorafenib. This establishes ferroptosis as an effective mechanism for inducing HCC cell death [13]. Therefore, triggering ferroptosis in HCC cells may be a viable strategy to enhance the efficacy of anticancer drugs.

The active compounds of traditional Chinese medicines are high efficacy and low toxicity. They enhance the effects of chemotherapy and targeted therapy, while also alleviating adverse reactions in patients and improving quality of life [14,15]. The primary active component of the Chinese herb *Cistanche deserticola* is echinacoside (ECH), which exhibits multiple functions, including antioxidant effects [16], improving cognitive impairment [17,18], anticancer activity [19,20,21,22,23,24], preventing osteoporosis [25], and providing neuroprotection [26,27,28]. Several studies have investigated the role of ECH and its mechanisms in the treatment of HCC. Previous research indicated that ECH exhibits anticancer effects in mouse models of HCC and HepG2 cells [29]. However, the precise mechanism by which ECH acts on ferroptosis in HCC remains unclear. Therefore, elucidating HCC pathogenesis from multiple dimensions is crucial for developing effective therapeutic agents.

To investigate the specific targets and mechanisms of ECH in HCC, this study comprehensively analyzed the potential targets and complex mechanisms of ECH treatment for HCC through network pharmacology, bioinformatic, molecular docking, and molecular dynamics simulations. It validated the binding affinity and stability of ECH with key targets. Furthermore, the HepG2 and Huh7 cell models were employed to experimentally validate these predictions. Our finding enriches the understanding of ECH’s pharmacological mechanisms, providing a scientific foundation for future pharmacological research, drug development, and clinical translation. The workflow chart is presented in graphical abstract.

## 2. Results

### 2.1. Target Collection

ECH’s chemical structure was shown in Figure 1A. ECH targets were identified through in silico screening in relevant databases, resulting in 473 unique targets after eliminating duplicates. To determine HCC-associated targets, we screened the OMIM and GeneCards databases, yielding a total of 9889 HCC targets. After finding the intersection between these two sets, we identified 183 ECH-treated HCC targets (Figure 1B). This result suggested that ECH played a significant role in HCC treatment.

Ferroptosis, a novel form of regulated cell death, shows considerable potential in HCC therapy. Therefore, we hypothesized that there were a close relationship between ECH’s anti-HCC mechanism and ferroptosis. Subsequently, we collected 1761 ferroptosis-related genes from the GeneCards and FerrDb. We identified 50 overlapping ferroptosis genes in ECH and HCC (Figure 1C). The results suggested a potential link between ECH’s ability to suppress HCC and its impact on ferroptosis.

### 2.2. PPI Network Analysis and KEGG, GO Analysis

The protein–protein interaction (PPI) network was constructed for ferroptosis-related genes in ECH-treated HCC (Figure 1D). The MCC algorithm was applied to calculate the centrality scores. The top 10 core genes identified by maximal clique centrality (MCC) were TP53, AKT1, EGFR, ESR1, SRC, JAK2, MMP2, STAT1, PPARG, and ANXA5 (Figure 1E). TP53 was identified as the primary core gene in ECH-mediated ferroptosis in HCC.

Further Kyoto Encyclopedia of Genes and Genomes (KEGG) and Gene Ontology (GO) pathway enrichment analyses were performed on ferroptosis targets in the ECH treatment of HCC. The results revealed that ECH treatment significantly enriched the ferroptosis signaling pathway among multiple key pathways (Figure 2A). GO enrichment analysis indicated that the TP53 signaling pathway was significantly enriched in molecular function (Figure 2B), playing a central role in the treatment of HCC.

### 2.3. Molecular Docking Result

Molecular docking is a powerful tool for predicting the binding affinity between active compounds and potential targets. Previous network pharmacology analysis identified TP53 as a core target for ECH-mediated ferroptosis in HCC. Molecular docking analysis was performed between ECH and TP53 (Figure 3). The docking results showed a binding energy of −8.9 kcal/mol for the interaction between ECH and TP53.Notably, TP53 showed the lowest docking energy among the 10 core targets identified by using network pharmacology and PPI analysis, suggesting it was ECH’s highest-affinity binding partner. Generally, a binding energy below −5 kcal/mol was indicated favorable binding affinity between receptor-ligand interaction [30]. Hydrogen bond interactions were observed between ECH and the TP53 protein amino acid residues GLU-16, GLU1591, GLU1543, ASN-1498, and HIS-68. Our findings indicated that ECH binds TP53 with significant affinity, and TP53 showed highest affinity among the core targets, supporting it was the key mediator of ECH-induced ferroptosis in HCC.

### 2.4. Molecular Dynamics Simulation Result

Additionally, molecular dynamics simulations were performed to evaluate the stability of the ECH-TP53 complex. The root mean square deviation (RMSD) of the ligand-protein complex reflects its overall conformational fluctuations during MD simulation; lower RMSD values indicate greater complex stability. The ECH-TP53 complex swiftly attained equilibrium and remained stable after approximately 50 ns, demonstrating minimal fluctuations (approximately 0.2–0.3 nm) throughout the simulation timescale and a high dynamic stability in the ECH-TP53 complex’s (Figure 4A). Root mean square fluctuation (RMSF) reflects the local flexibility of protein residues. The most TP53 residues exhibited low fluctuations (RMSF < 0.3 nm), indicating good rigidity (Figure 4B). The radius of gyration (Rg) remained stable around 2.0 nm throughout the simulation, suggesting that binding to ECH keeps TP53 compact (Figure 4C).

Solvent-accessible surface area (SASA) quantifies the surface area of the protein exposed to solvent. The SASA of the complex remained stable throughout the simulation, indicating that the TP53 hydrophobic core stayed well protected after ligand binding and the protein maintained a compact and stable structure (Figure 4D). A stable and persistent hydrogen bond network formed between ECH and the TP53 protein during the simulation (Figure 4E). The free energy landscape plot revealed the thermodynamic stability of the system across different conformational states. There is a distinct and concentrated low energy basin (blue region), corresponding to conformational states with RMSD values between 0.2 and 0.3 nm, and Rg values between 2.0 and 2.5 nm (Figure 4F). The ECH-TP53 complex predominantly sampled the lowest-energy, most stable conformation during the simulation.

### 2.5. ECH Inhibits HCC Cell Growth

Cell viability of HepG2 and Huh7 cells treated with ECH was measured using the cell counting kit-8 (CCK-8) assay. ECH was found to have significant inhibitory effects on both cell lines (Figure 5A,B). After 24 h, increasing concentrations of ECH significantly reduced cell viability in both cell lines compared with control. Using GraphPad Prism, the half-maximal inhibitory concentration (IC_50_) after 24 h were 291.2 μmol/L (HepG2) and 387.6 μmol/L (Huh7). Notably, LO2 cell viability showed no significant inhibition at the same concentrations (Figure 5C). Consequently, HepG2 cells were treated with 100, 200, 300 μmol/L and Huh7 cells with 100, 200, 400 μmol/L for further investigation. To confirm the observed cell death caused by ferroptosis, we employed ferrostatin-1 (Fer-1), a widely used and specific ferroptosis inhibitor (Appendix A). We found a non-cytotoxic concentration of Fer-1 for subsequent co-treatment experiments. Overall, our results demonstrated that ECH was significantly inhibited the HCC cell growth.

### 2.6. ECH Induces Ferroptosis in HCC Cells

To investigate whether ECH induces ferroptosis in HCC cells. Ferroptosis was primarily included iron-dependent lipid peroxidation (LPO) and iron accumulation [9]. Thus, we assessed intracellular LPO levels using fluorescence inverted microscopy. The results showed that ECH was significantly accumulated LPO in both HepG2 and Huh7 cells (Figure 6A). The intracellular Fe^2+^ levels was measured by using a ferrous ion assay kit, the results found that ECH induced intracellular Fe^2+^ accumulation in HCC cells (Figure 6B), while co-treatment with Fer-1 and ECH reversed this trend. As glutathione (GSH) depletion and malondialdehyde (MDA) accumulation are key factors for ferroptosis, our results found a reduced GSH levels in HCC cells (Figure 6C) and an increased MDA levels (Figure 6D), and co-treatment with Fer-1 and ECH reversed these trends, suggesting ferroptosis mediated ECH-induced growth inhibition in HCC cells.

### 2.7. ECH Induces Ferroptosis in HCC Cells via the TP53/SLC7A11/GPX4 Signaling Pathway

To further investigate TP53’s role in HCC, we performed overall survival analysis using the survival package in R software on TCGA-LIHC clinical data, comparing the expression levels in HCC and normal tissues. The results were visualized via Kaplan–Meier univariate survival analysis plots. As shown in Figure 7A, TP53 expression was significantly differentiated between high-risk and low-risk groups, HCC patients with poor prognosis were observed a low TP53 expression. TP53 serves as a key regulator in the solute carrier family 7 member 11(SLC7A11)/glutathione peroxidase 4(GPX4) signaling pathway, and promote ferroptosis. By suppressing SLC7A11 expression, TP53 inhibits cystine uptake, and reducing GPX4 levels. This decrease in cellular antioxidant capacity leads to elevated LPO, triggering ferroptosis and inhibiting tumor progression [31,32]. Therefore, we evaluated the expression of TP53,SLC7A11 and GPX4 in ECH-treated HCC cells. TP53 mRNA was upregulated, whereas SLC7A11 and GPX4 were downregulated (Figure 7B). Co-treatment with Fer-1 and ECH reversed these trends. Furthermore, the TP53/SLC7A11/GPX4 expression levels elicited an increased TP53 expression, and a decreasing SLC7A11 and GPX4 expression. This upregulation and downregulation trend was also reversed by combined Fer-1 and ECH intervention (Figure 7C and 7D). The result suggested that ECH could suppress the SLC7A11/GPX4 pathway and promote ferroptosis in HCC cells.

## 3. Discussion

Liver cancer is a common malignant tumor that poses a serious threat to human health. Among primary liver cancers, 80–90% are pathologically classified as HCC [1]. China poses the highest HCC incidences, prevalences, and mortality rates of HCC in the world, underscoring the urgent need for effective therapeutic interventions. ECH is a medicinal herb with dual food and medicinal properties. Previous studies demonstrated that it is non-toxic or exhibits minimal side effects, and recognized as a safe, natural compound [33]. ECH exhibits inhibitory effects against multiple malignant cancers, including lung cancer [34], breast cancer [35], ovarian cancer [20], and HCC [36]. The clinical safety and efficacy of ECH provide natural advantages in tumor therapy, making it a promising candidate for HCC clinical treatment. Other natural compounds, like Matrine, were also inhibited the proliferation of HCC cells and induced apoptosis by suppressing the AKT/GSK3β/β-catenin signaling pathway and reducing β-catenin transcriptional activity [37]. The ginsenoside CK induced ferroptosis in HCC cells by inhibiting FOXO1 phosphorylation, thereby downregulating SLC7A11 and GPX4 expression to suppress HCC [38]. These studies suggest that HCC pathogenesis is closely associated with various forms of cell death, including necrosis, apoptosis, autophagy, and ferroptosis. These finding suggest that cell death could serve as a indicator for anticancer drug development. However, fewer studies have been conducted to determine which form of cell death predominated in HCC. Notably, prior study has demonstrated that ECH can inhibit tumor cell proliferation and migration at the cellular level by regulating apoptotic pathways [39]. Ferroptosis, a relatively novel form of regulated cell death, is increasingly recognized for its pathophysiological effects in various diseases. In cancer, it is believed as a promising therapeutic vulnerability, specially in drug-resistant and metastatic cancers [40]. HCC cells exhibit higher expression of RAS oncogenes and elevated Fe^2+^ content suggesting an increased sensitivity to ferroptosis [11]. In addition, microRNA-214-3p could also promote ferroptosis in HCC cells by targeting and suppressing ATF4 expression [41]. Therefore, our study explores the molecular mechanisms underlying ECH-induced ferroptosis in HCC, with a focus on a key pathway in this process.

Our study represents several novel findings. Firstly, we demonstrated that ECH has anti-HCC properties, which may be contributed to its ability of inducing ferroptosis. Notably, HepG2 cells are derived from hepatoblastoma rather than typical adult HCC, the key phenotypes of ECH-induced ferroptosis and its regulatory effects on the TP53/SLC7A11/GPX4 pathway were consistent in Huh7 cells, ensuring the applicability of our conclusions to HCC. Secondly, ECH modulates multiple pathways in HCC treatment, with the TP53 pathway showing a significant enrichment. Intersection analysis of ECH, HCC, and ferroptosis targets were further confirmed TP53 serves as the first core gene regulated by ECH in HCC ferroptosis. Notably, HCC patients with elevated TP53 expression showed favorable outcomes. Molecular docking analysis revealed an good binding affinity between ECH and TP53, and molecular dynamics simulations confirmed the stable binding of the ECH-TP53 complex. These findings suggest that TP53 plays a crucial role in the treatment of HCC with ECH. To experimentally validate these hypothesis and assess the therapeutic outcome of ECH in HCC, we evaluated its antitumor activity in two widely used HCC cell lines, HepG2 and Huh7 cells [42,43]. and the non-tumorigenic LO2 hepatocyte line was used to evaluate toxic effects. Our finding confirmed ECH could activate TP53/SLC7A11/GPX4 pathway to trigger ferroptosis in HCC cells. These findings suggest that TP53 plays a central role in the treatment of HCC with ECH.

TP53 could suppress tumor progression by inhibiting GPX4 level through the downregulation of SLC7A11 expression, followed by inducing ferroptosis [44]. The SLC7A11/GPX4 pathway is classic for regulating ferroptosis and is closely associated with tumor progression. In tumor cells, TP53 activation suppresses SLC7A11 expression and induces ferroptosis, thereby inhibiting tumor cells’ proliferation and survival. SLC7A11 is a subunit of the cystine/glutamate antiporter system (system xCT), which functions as a heterodimer within the phospholipid bilayer of the cell membrane. xCT belongs to the amino acid transporter family and comprises two subunits. The light chain of SLC7A11 functions as a transporter, primarily facilitates an exchange ratio of 1:1 of extracellular cystine for intracellular glutamate. The heavy chain subunit (SLC3A2) acts as a chaperone protein solely involved in maintaining the stability of SLC7A11. SLC7A11 could effectively inhibit the development of HCC [45]. In wild-type TP53, SLC7A11 has been confirmed as a target of TP53-mediated transcriptional repression; however, TP53 mutants fail to inhibit SLC7A11. Kim et al. [46] demonstrated that wild-type TP53 is present in human HCC cells with low mutation frequency [31,47]. GPX4 is a primary inhibitor of ferroptosis, an essential enzyme for clearing lipoxygenative free radicals. GPX4 utilizes the co-factor GSH to convert toxic lipid peroxyl radicals (PLOOHs) into non-toxic lipooxygenyl radicals (PLOHs). Decreased or absent GPX4 expression leads to the accumulation of massive amounts of lipid peroxide, disrupting membrane integrity and inducing cell death [48]. Network pharmacological and bioinformatic analyses predicted that ECH induces ferroptosis in HCC by modulating the TP53/SLC7A11/GPX4 pathway (upregulating TP53 and downregulating SLC7A11/GPX4). Crucially, this effect was reversed by Fer-1, confirming a ferroptosis-dependent mechanism.

Although predictive and experimental evidence suggests that ECH targets the TP53/SLC7A11/GPX4 pathway to promote ferroptosis and inhibit HCC progression, this study has several limitations. First, despite identifying multiple pathways in silico, we focused only on the TP53 axis, necessitating future exploration of other targets. Second, direct empirical validation of the ECH-TP53 interaction is still lacking. Third, the TP53 dependence of this ferroptosis induction remains to be confirmed through knockout or inhibition models. Fourth, the observation that ECH induces ferroptosis in HuH7 cells (harboring the R249S TP53 mutation) suggests a potential ability to modulate mutant TP53 function, possibly by binding to conserved residues such as GLU173 and ASN239. This reactivation hypothesis warrants future structural and functional investigation.

Although predictive and experimental evidence suggests that ECH can target the TP53/SLC7A11/GPX4 pathways to promote ferroptosis and inhibit HCC progression, this study has several limitations. First, despite identifing multiple pathways in silico, our study focused only on the TP53 pathway, necessitating exploration of other potential targets. Second, direct empirical validation for the ECH-TP53 interaction is still lacking. Third, the TP53 dependence of this ferroptosis induction remain to be indentified, by using TP53 knockout/inhibition cell models. Fourth, the observation that ECH induces ferroptosis in HuH7 cells (harboring the R249S TP53 mutation) suggests a potential ability to modulate mutant TP53 function, possibly by binding to conserved residues such as GLU173 and ASN239. This reactivation hypothesis warrants future structural and functional investigation. Furthermore, ECH’s clinical translation is hindered by significant pharmacokinetic challenges. As a glycoside with an ester bond, it exhibits low oral bioavailability (approx. 0.83%) [49], primarily due to poor intestinal absorption, enzymatic and microbial hydrolysis, and extensive first-pass metabolism [50]. Future studies should therefore prioritize formulation strategies, such as phospholipid complexes, nanomedicines, or liposomes, to enhance stability, absorption, and systemic exposure.

## 4. Materials and Methods

### 4.1. In Silico Screening of Potential Therapeutic Targets for ECH

We retrieved the SMILES structure of ECH from the PubChem (https://pubchem.ncbi.nlm.nih.gov/, accessed on 10 August 2025), and imported the ECH SMILES file from three databases: SwissTargetPrediction Database (http://swisstargetprediction.ch, accessed on 11 August 2025), Pharmmapper Database (http://www.lilab-ecust.cn/pharmmapper/, accessed on 10 August 2025), and Comparative Toxicogenomic Database (CTD) (https://ctdbase.org/, accessed on 9 August 2025). ECH targets were collected from literature in the PubMed (https://pubmed.ncbi.nlm.nih.gov/, accessed on 8 August 2025). The retrieved targets were standardized in the UniProt database. Finally, we combined the target information, removed duplicates, and merged the data to obtain a unified set of potential therapeutic targets for ECH.

### 4.2. In Silico Screening of Potential Disease Targets

Searches were conducted in the GeneCards (https://www.genecards.org, accessed on 12 August 2025) [51] and the OMIM (https://www.omim.org, accessed on 12 August 2025) [52] using the keyword “hepatocellular carcinoma” to identify potential HCC-related targets. The inclusion criteria for HCC targets in silico screening were set as “protein coding” in both databases. Visualization was performed using VENNY 2.1 (https://bioinfogp.cnb.csic.es/tools/venny/, accessed on 14 August 2025) with the inclusion criteria set to “protein coding”.

### 4.3. Sources of Ferroptosis Targets

The keyword “ferroptosis” was used to collect ferroptosis target genes from the GeneCards (https://www.genecards.org/, accessed on 13 August 2025) and the FerrDb (http://www.zhounan.org/ferrdb/current/, accessed on 13 August 2025). After removing duplicates, the intersection of the two sets was taken to obtain the final set of ferroptosis targets. The inclusion criteria for in silico screening were set as follows: the target was required to be a driver, suppressor, marker, or unclassified gene in FerrDb, and a protein-coding gene in GeneCards.

### 4.4. PPI Network Analysis

The STRING (https://cn.string-db.org/, accessed on 15 August 2025) was used for PPI network analysis [53]. Targets for ECH, HCC and ferroptosis were integrated into the STRING. The organism was set as ‘*Homo sapiens*’, and the maximum confidence threshold was set to 0.7 to obtain the TSV file for constructing the PPI network. Subsequently, Cytoscape 3.9.0 software was used for further topological analyses. Node size, shape, color, and layout were adjusted to visualize the constructed PPI network. The MCC algorithm was applied to calculate the centrality scores and identify the top 10 core genes.

### 4.5. GO and KEGG Enrichment Analysis

The potential targets of ECH therapy for HCC were imported into the DAVID (https://davidbioinformatics.nih.gov/, accessed on 15 August 2025) to perform GO and KEGG pathway enrichment analysis. A *p*-value ≤ 0.05 was significant differences.

### 4.6. Molecular Docking

We retrieved the ECH structure from the PubChem and converted SDF to Protein Data Bank format by using OpenBabel 3.1.1. The crystal structure of human TP53 (PDB ID: 1TSR) was retrieved from the RCSB Protein Data Bank (https://www.rcsb.org, accessed on 16 August 2025) [54]. The ECH structure file was subsequently preprocessed by Auto Dock Vina 1.2.7 [55], and the docking results were visualized and output using PyMOL 2.6 [56].

### 4.7. Molecular Dynamics Simulation

To validate the rationality and stability of the docking results between small molecules and 3D protein structures, molecular dynamics simulations of protein-ligand complexes were performed using GROMACS 2023.2 under isothermal and isobaric conditions. The CHARMM36 force field was employed for protein processing, while the Gaff2 force field was used for small molecules. The topology files were prepared for the protein and ligand, followed by the construction of the complex topology file. The complex was then hydroxylated using the TIP3P water model, and a cubic box was constructed that extended 1.0 nm beyond the protein boundary. Appropriate ionic simulation system charges were also added. The solvent system consisted water with NaCl for charge neutralization. The parameters were as follows: temperature 310 K and pressure 1 bar. Each phase ran for 100 ps, with trajectories saved every 5 ps. DuIvy Tools v0.5.0 (https://duivytools.readthedocs.io/en/latest/DIT_old.html, accessed on 20 August 2025) was then used to calculate and visualize the following protein metrics: RMSD, RMSF, Rg, and SASA.

### 4.8. Reagents and Antibodies

ECH (catalog no. B21209, purity ≥ 98%) was purchased from Shanghai Yuanye Bio-Technology Co., Ltd. (Shanghai, China). Fer-1 and CCK-8 were purchased from MedChemExpress (Monmouth Junction, NJ, USA). Erastin was purchased from Selleck Chemicals (Houston, TX, USA). The TB Green^®^ Premix Ex Taq™ II (Tli RNaseH Plus), GAPDH, TP53, SLC7A11, and GPX4 primers were synthesized by Sangon Biotech Co., Ltd. (Shanghai, China). All primer information is documented in Appendix A. Antibodies targeting GAPDH, TP53, SLC7A11, and GPX4 were provided by Proteintech (Wuhan, China).

### 4.9. In Vitro Experimental Validation

#### 4.9.1. Cell Culture

The human normal hepatocytes cell line LO2, human hepatoblastoma cell line HepG2, and human hepatocellular carcinoma (HCC) cell line HuH7 were obtained from Procell (Wuhan, China). The authenticity of all employed cell lines was ensured through short tandem repeat (STR) profiling. All cells were cultured in high-glucose DMEM containing 10% fetal bovine serum (Procell, Wuhan, China). All cells were maintained in a 37 °C, 5% CO_2_ incubator and the culture medium was changed every 2–3 days.

#### 4.9.2. CCK-8 Assay

The LO2, HepG2, and Huh7 cells of each group were collected and seeded in 96-well plates within 8 × 10^3^ cells/well for culturing overnight. Then, the cells were treated with different concentrations of ECH. The control group was treated with 0.1% DMSO and the blank group was treated with a cell-free medium. After 24 h, the supernatant was removed and the cells were washed twice with PBS to eliminate its effect on absorbance. Fresh medium containing 10% CCK-8 working solution was added to each well, which were incubated at 37 °C for 1 h. Finally, the absorbance at 450 nm was quantified using a microplate reader (Thermo Multiskan Sky, Waltham, MA, USA).

#### 4.9.3. Ferroptosis-Related Indicator Detection

The MDA assay kit (SABC, Beijing, China), the GSH assay kit (SABC, Beijing, China), the C11-BODIPY 581/591 probe (Thermo Fisher Scientific, Waltham, MA, USA), and the Fe^2+^ assay kit (Beyotime, Shanghai, China) were employed to measure the levels of MDA, GSH, LPO, and Fe^2+^. For LPO detection, HepG2 and Huh7 cells (3 × 10^5^ cells/well) were seeded into 6-well plates and treated with the relevant drugs. The cells were then stained for LPO using DMEM medium containing 10 mM BODIPY 581/591 C11. After Hoechst 3342 staining of live cell nuclei, the samples were incubated at 37 °C in a cell culture incubator for 30 min. The cells were then washed twice with PBS and imaged using an inverted fluorescence microscope (Nikon Ts2R, Tokyo, Japan). Images were analyzed using Image J v1.8.0.345.

#### 4.9.4. Quantitative Real-Time Polymerase Chain Reaction (qRT-PCR)

Total RNA was extracted using an RNA extraction kit. Subsequently, the cDNA was synthesized by reverse transcription with the PrimeScript™ RT Reagent Kit under the following conditions: 37 °C for 15 min and 85 °C for 5 s. The cDNA was then amplified using TB Green^®^ Premix Ex Taq™ II (Tli RNaseH Plus) with the following program: pre-denaturation at 95 °C for 30 s, followed by 40 cycles of 95 °C for 5 s and 60 °C for 34 s. All primer sequences used in the qRT-PCR assay were provided in Appendix A and were designed by Sangon Biotechnology (Shanghai, China).

#### 4.9.5. Western Blotting (WB)

Protein expression levels were detected through WB analyses. Cell samples were lysed using a radio immunoprecipitation assay (RIPA) buffer containing phenylmethanesulfonyl fluoride (PMSF), ethylene diamine tetraacetie acid (EDTA), and protease inhibitors. Protein content was detected using a bicinchoninic acid (BCA) protein assay kit. The protein samples were separated using sodium dodecyl sulfate polyacrylamide gel electrophoresis (SDS-PAGE) and transferred to polyvinylidene fluoride (PVDF) membranes. The membranes were then blocked for 2 h in Tris-buffered saline with Tween-20 (TBST) containing 5% non-fat dry milk and 0.05% Tween-20. After three washes with TBST, the membranes were incubated overnight at 4 °C with the following antibodies: GAPDH (1:5000), TP53 (1:10,000), SLC7A11 (1:2000), and GPX4 (1:5000). The membrane was washed with TBST three times and incubated with the secondary antibody at room temperature for 1 h. After three additional TBST washes, the ECL chemiluminescent solution (Biosharp, Hefei, China) was added and the membrane was visualized using the ChemiDoc XRS system (Bio-Rad, Hercules, CA, USA).

### 4.10. Statistical Analysis

All statistical analyses were performed using GraphPad Prism 8.0.2, and all data were presented as mean ± standard. The normality of the data and the homogeneity of the variance were assessed using the Shapiro-Wilk and Brown-Forsythe tests. The results confirmed normal distribution and homogeneity of variance. Significant differences among multiple groups were further analyzed using one-way ANOVA, with *p* < 0.05 considered as statistically significant (* indicates *p* < 0.05, ** indicates *p* < 0.01, *** indicates *p* < 0.001, **** indicates *p* < 0.0001).

## 5. Conclusions

Our findings indicate that ECH exerts its anti-HCC effect by activating the TP53/SLC7A11/GPX4 pathway, specifically through increasing TP53 while inhibiting SLC7A11 and GPX4 expression. This cascade elevates LPO in HCC cells, thereby trigger them to ferroptosis and effectively inhibiting tumor progression. Furthermore, the results imply the potential therapeutic targets for ECH in treating HCC, offering a theoretical basis for future clinical translation.

## Figures and Tables

**Figure 1 ijms-27-00411-f001:**
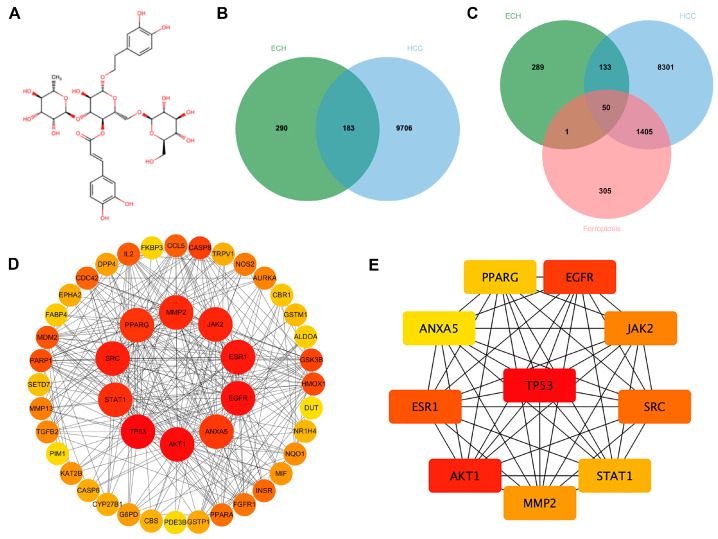
Network pharmacology analysis of ECH treatment for HCC. (**A**) The chemical structure of ECH. (**B**) Intersection of ECH and HCC targets. (**C**) Intersection of ECH, HCC, and ferroptosis targets. (**D**) PPI network analysis of the intersection of ECH, HCC, and ferroptosis targets, with the inner circle showing the top ten core targets for ECH-mediated ferroptosis in HCC. (**E**) The top ten core targets for ECH-mediated regulation of HCC ferroptosis, with TP53 as the primary core target. Abbreviations: ECH: echinacoside. HCC: hepatocellular carcinoma. PPI: protein–protein interaction.

**Figure 2 ijms-27-00411-f002:**
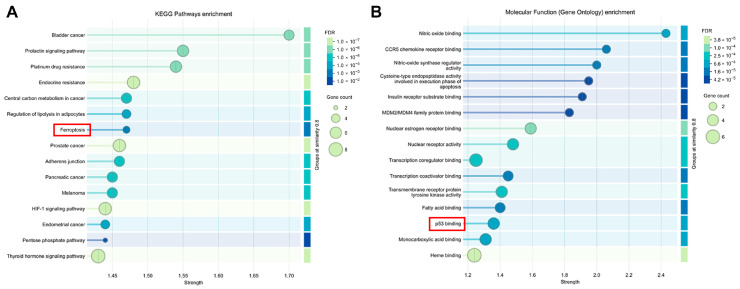
Functional enrichment analyses of ECH against HCC. (**A**) Lollipop chart of KEGG analysis for ECH treatment of HCC. (**B**) Lollipop chart showing the MF section of the GO analysis for ECH treatment of HCC. Abbreviations: KEGG: Kyoto Encyclopedia of Genes and Genomes. ECH: echinacoside. HCC: hepatocellular carcinoma. MF: molecular function. GO: Gene Ontology.

**Figure 3 ijms-27-00411-f003:**
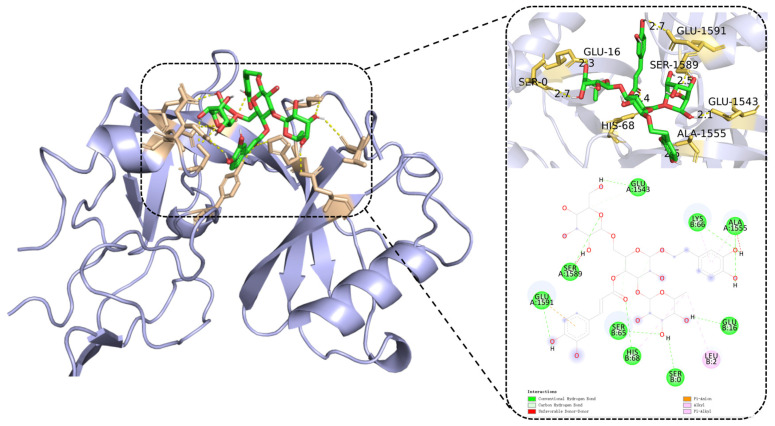
Molecular docking results of ECH with the core target TP53. The docking binding energy between ECH and TP53 is −8.9 kcal/mol, indicating a stable interaction, green represents carbon atom bonds, red represents the oxygen atom. Abbreviations: TP53: tumor protein 53. ECH: echinacoside. GLU: glutamic acid. ASN: Asparagine. HIS: Histidine.

**Figure 4 ijms-27-00411-f004:**
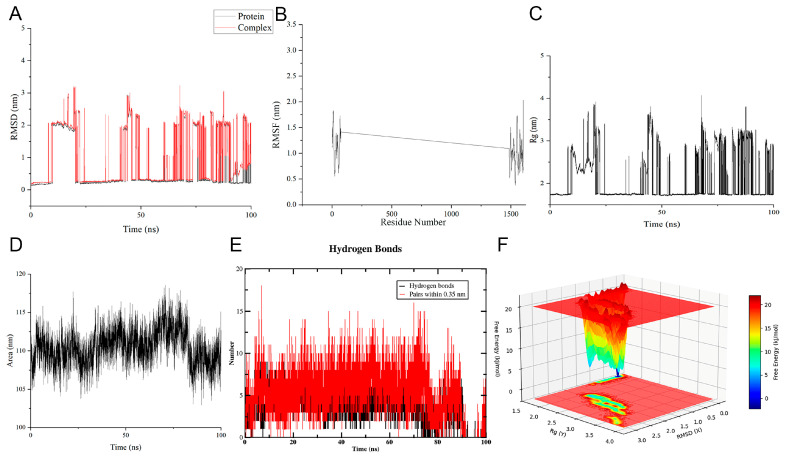
Results of molecular dynamics simulation of the TP53-ECH complex. (**A**) RMSD; (**B**) RMSF; (**C**) Rg; (**D**) SASA; (**E**) number of hydrogen bonds; (**F**) free energy landscape plot. Abbreviations: TP53: tumor protein 53. ECH: echinacoside. RMSD: root mean square deviation. RMSF: root mean square fluctuation. Rg: radius of gyration. SASA: solvent-accessible surface area.

**Figure 5 ijms-27-00411-f005:**
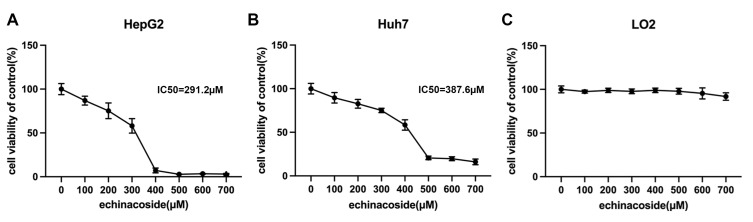
The results of HCC cell growth inhibited by ECH. (**A**) HepG2 cells were treated with different concentrations of ECH for 24 h. (**B**) Huh7 cells were treated with different concentrations of ECH for 24 h. (**C**) LO2 cells were treated with different concentrations of ECH for 24 h. Abbreviations: ECH: echinacoside. HCC: hepatocellular carcinoma. IC_50_: half-maximal inhibitory concentration.

**Figure 6 ijms-27-00411-f006:**
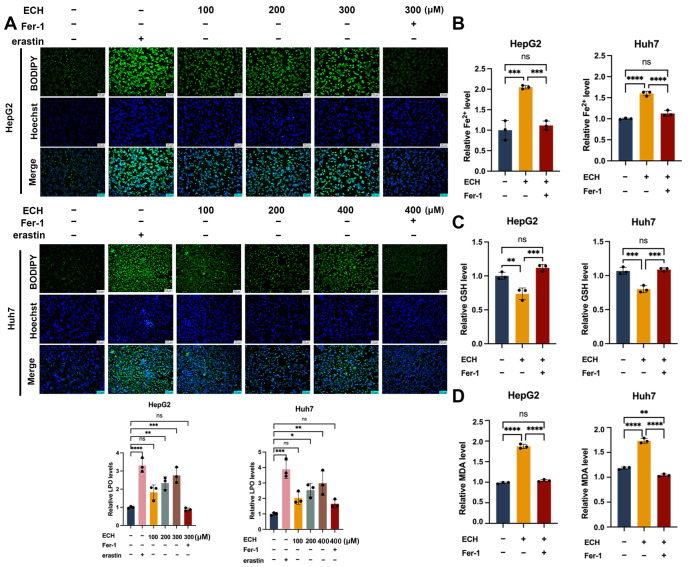
Ferroptosis in HCC cells induced by ECH. (**A**) LPO levels in HepG2 and Huh7 cells treated with erastin (10 μmol/L), Fer-1 (2 μmol/L), and ECH for 24 h were detected and quantified using fluorescence inverted microscopy. (**B**) Fe^2+^ levels after treated with ECH (300 µmol/L) and Fer-1 (2 µmol/L) for 24 h in HepG2 and Huh7 cells. (**C**) GSH levels after treated with ECH (300 µmol/L) and Fer-1 (2 µmol/L) for 24 h in HepG2 and Huh7 cells. (**D**) MDA levels after treated with ECH (300 µmol/L) and Fer-1 (2 µmol/L) for 24 h in HepG2 and Huh7 cells. The plus sign (+) indicates drug intervention, and the minus sign (−) indicates absence of the drug. Data were represented as mean ± SD (*n* = 3); ns = no significance, * *p* < 0.05, ** *p* < 0.01, *** *p* < 0.001, and **** *p* < 0.0001 versus control group. Abbreviations: ECH: echinacoside. HCC: hepatocellular carcinoma. Fer-1: ferrostatin-1. GSH: glutathione. MDA: malondialdehyde.

**Figure 7 ijms-27-00411-f007:**
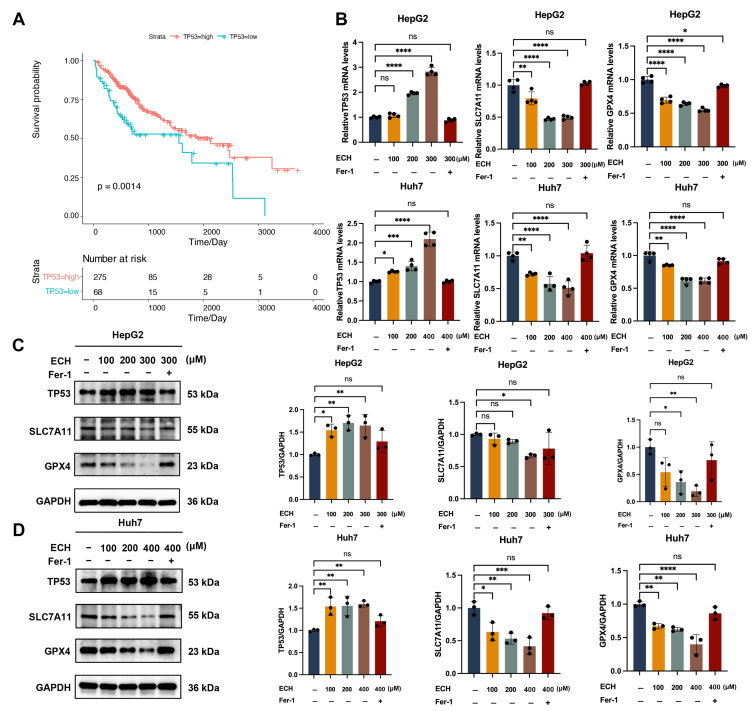
Ferroptosis induced by ECH in HCC cells via the TP53/SLC7A11/GPX4 signaling pathway. (**A**) Kaplan-Meier univariate survival analysis of TP53 expression in HCC versus normal tissues. (**B**) The TP53, SLC7A11, and GPX4 mRNA levels in HepG2 and Huh7 cells following a 24 h treatment with various concentrations of ECH and Fer-1 (2 μmol/L). (**C**) The TP53, SLC7A11, and GPX4 protein expression levels in HepG2 cells after 24 h treatment with different concentrations of ECH and Fer-1 (2 μmol/L). (**D**) The protein expression levels of TP53, SLC7A11, and GPX4 in Huh7 cells after treatment with different concentrations of ECH and Fer-1 (2 μmol/L) for 24 h. Data are represented as mean ± SD (*n* = 3); ns = no significance, * *p* < 0.05, ** *p* < 0.01, *** *p* < 0.001, and **** *p* < 0.0001 vs. control group. Abbreviations: ECH: echinacoside. HCC: hepatocellular carcinoma. TP53: tumor protein 53. Fer-1: ferrostatin-1. SLC7A11: solute carrier family 7 member 11. GPX4: glutathione peroxidase 4.

## Data Availability

No new data were created or analyzed in this study. Data sharing is not applicable to this article.

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
