# Peer review of "Echinacoside as a Novel Ferroptosis Inducer in Hepatocellular Carcinoma: Mechanistic Insights from TP53/SLC7A11/GPX4 Pathway Modulation"

_ijms, 2025, doi:10.3390/ijms27010411_

Round 1
Reviewer 1 Report
Comments and Suggestions for Authors
This manuscript utilizes methods such as network pharmacology, molecular docking, and cellular-level validation to demonstrate that echinacoside acts as a novel ferroptosis inducer by modulating the TP53/SLC7A11/GPX4 pathway, thus has potential of anti-Hepatocellular Carcinoma. After reviewing this manuscript, I have the following questions that require clarification from the author:
- The manuscript frequently uses the term "screening," such as in lines 334 and 344, et al. This expression lacks precision. Could the authors validate the interaction between Echinacoside and TP53 through ligand-target protein interaction experiments? If such experimental evidence cannot be provided, my recommendation is to replace "screening" with "virtual screening" or "in silico screening."
- In the cell experiments, three cell lines,LO2, HepG2, and Huh7,were used. Is it possible to standardize them to a single cell line, such as HepG2, which is a typical hepatocellular carcinoma cell line? Therefore, the authors need to provide necessary explanations in the manuscript regarding why these three cell lines were selected.
- In line 165, the phrase "whose chemical structure is shown in Figure 5A" is ambiguous in its reference. If it refers to the chemical structure of echinacoside, as a key subject of study, its chemical structure should not appear as late as in Figure 5.
- ”Ferroptosis, a form of iron-dependent cell death that was discovered by Dixon's team in 2012“, is a statement that directly copies from previous literature and lacks professionalism. Dixon is merely the first author of the paper, not the team leader responsible for the article. This requires correction.
- In the KEGG enrichment analysis, ferroptosis appears as a significantly enriched pathway. However, among the displayed overlapping targets, aside from TP53, no other ferroptosis-related targets are visible. Could you extract all the targets enriched in this pathway?
- Some writing habit errors and grammatical errors require careful revision. For example, in line 174, the abrupt mention of "Fer-1" is problematic, as its explanation appears only in a subsequent sentence. Other issues also need to be thoroughly reviewed by the author in the manuscrpt.
Some grammatical errors should be corrected, such as "HepG2 and Huh7 cells to further validate the findings", it is not a standard expression.
Author Response
|
Dear Reviewer, We sincerely appreciate the reviewer’s valuable time and effort in evaluating our manuscript. Thank you very much for your careful review and constructive comments on our manuscript entitled “Echinacoside as a Novel Ferroptosis Inducer in Hepatocellular Carcinoma: Mechanistic Insights from TP53/SLC7A11/GPX4 Pathway Modulation” (Manuscript ID: ISSN 1422-0067). Your insightful suggestions have been invaluable in helping us improve the quality and clarity of our work. We have revised the manuscript point-by-point according to your comments. All changes in the revised manuscript have been highlighted in red for your convenience. Our detailed responses are as follows.
|
|
2. Point-by-point response to Comments and Suggestions for Authors |
|
Comments 1: The manuscript frequently uses the term "screening," such as in lines 334 and 344, et al. This expression lacks precision. Could the authors validate the interaction between Echinacoside and TP53 through ligand-target protein interaction experiments? If such experimental evidence cannot be provided, my recommendation is to replace "screening" with "virtual screening" or "in silico screening."
|
|
Response 1: We sincerely thank you for your insightful and constructive comment. You have correctly pointed out that the term “screening” was imprecise in the context of our computational work. We have thoroughly revised the manuscript and replaced all instances of the imprecise “screening” with “in silico screening” (in lines 82,335,338,365,375, 379, 386) to accurately reflect our methodology. All changes to the text are displayed in red for easy identification.
|
|
Comments 2: In the cell experiments, three cell lines,LO2, HepG2, and Huh7,were used. Is it possible to standardize them to a single cell line, such as HepG2, which is a typical hepatocellular carcinoma cell line? Therefore, the authors need to provide necessary explanations in the manuscript regarding why these three cell lines were selected. |
|
Response 2: We thank you for raising this important point about our experimental design. We sincerely thank you for this valuable question regarding our choice of cell lines. We agree that HepG2 is a classic and widely used HCC cell line. The selection of these three cell lines was deliberate and based on the following scientific rationale, HepG2: As the you noted, it is a widely used, well-characterized TP53 wild-type hepatocellular carcinoma cell line, serving as our primary model. Huh7: This is another widely used HCC cell line. Our inclusion of both HepG2 and Huh7 was a deliberate design choice based on their established roles in HCC research and their complementary value for mechanistic insight. Notably, the concurrent use of HepG2 and Huh7 is common in high-impact studies on ferroptosis in HCC, such as in investigations of Donafenib ("Donafenib activates the p53 signaling pathway in hepatocellular carcinoma, induces ferroptosis, and enhances cell apoptosis[1]") and Aspirin ("Aspirin triggers ferroptosis in hepatocellular carcinoma cells through restricting NF-κB p65-activated SLC7A11 transcription[2]"). LO2: This is a non-tumorigenic human hepatocyte cell line. It was used to evaluate the potential selective toxicity of ECH between normal and cancerous liver cells, which is a crucial aspect for evaluating its therapeutic potential. The lower cytotoxicity of ECH in LO2 cells compared to HepG2 and Huh7 cells strengthens its promise as a selective anti-HCC agent. Furthermore, as you recommended, we have included a justification in the Discussion section to contextualize this experimental design for the reader. All newly added text has been highlighted in red for your easy review (in lines 299-304). The relevant supporting references have been incorporated accordingly. We believe these revisions directly address your concern, provide full transparency for our experimental design, and enhance the overall clarity and rigor of the manuscript. Thank you once again for this constructive suggestion.
Comments 3: In line 165, the phrase "whose chemical structure is shown in Figure 5A" is ambiguous in its reference. If it refers to the chemical structure of echinacoside, as a key subject of study, its chemical structure should not appear as late as in Figure 5. Response 3: We are very grateful to you for identifying this ambiguity and logical inconsistency. Your point is completely valid; the chemical structure of the core investigational compound, Echinacoside (ECH), should indeed be presented early in the manuscript for clarity. To address this, we have moved the chemical structure of ECH from the original Figure 5A to a new figure, now presented as Figure 2A, where it provides essential visual context at an appropriate stage of the Results section. Consequently, all subsequent figure numbers have been updated throughout the manuscript. Importantly, all related in-text citations (including those in lines 94 and 188) and figure legends have been carefully revised and are now correctly referenced. All these changes have been highlighted in red in the revised manuscript for your easy review. We sincerely appreciate your meticulous attention, which has significantly improved the logical flow of our article.
Comments 4: “Ferroptosis, a form of iron-dependent cell death that was discovered by Dixon's team in 2012", is a statement that directly copies from previous literature and lacks professionalism. Dixon is merely the first author of the paper, not the team leader responsible for the article. This requires correction. Response 4: We sincerely and profoundly apologize for this oversight and the unprofessional manner of citation in our original manuscript. We fully acknowledge that this was an error in academic rigor, and we are truly grateful to you for your careful reading and precise correction. It has been an important lesson for us.We have thoroughly revised this problematic statement and conducted a full-text check to ensure no similar issues remain. Specifically, in lines 42-43 of the revised manuscript, the description has been corrected to: “Ferroptosis, a form of iron-dependent regulated cell death, was first formally described by Dixon et al. in 2012.” This change, along with any other related adjustments, has been highlighted in red for your convenience. Thank you once again for your invaluable guidance, which has directly enhanced the accuracy and professionalism of our work. We greatly appreciate the time and expertise you have dedicated to improving our manuscript.
Comments 5: In the KEGG enrichment analysis, ferroptosis appears as a significantly enriched pathway. However, among the displayed overlapping targets, aside from TP53, no other ferroptosis-related targets are visible. Could you extract all the targets enriched in this pathway? Response 5: We sincerely thank you for this critical and insightful observation, which has highlighted an important limitation in our initial data presentation. You are right. In our original KEGG enrichment analysis figure, we only displayed the top ten most significantly enriched pathways for clarity, which inadvertently omitted the full list of specific targets within the ferroptosis pathway. We apologize for this lack of detail, which may have obscured the complete picture of our network pharmacology predictions. To fully address your concern and ensure complete transparency, we have now extracted and listed all targets from our candidate set that are annotated in the KEGG ferroptosis pathway. This list includes HMOX1, TP53, and TF. Thank you once again for your meticulous review, which has greatly enhanced the rigor and completeness of our work.
Comments 6: Some writing habit errors and grammatical errors require careful revision. For example, in line 174, the abrupt mention of "Fer-1" is problematic, as its explanation appears only in a subsequent sentence. Other issues also need to be thoroughly reviewed by the author in the manuscript. Response 6: We deeply appreciate your meticulous reading and for pointing out the abrupt introduction of “Fer-1.” This is a valid critique, and we sincerely apologize for the lack of clarity. To address this, we have conducted a comprehensive revision of the entire manuscript to correct this and other grammatical or stylistic issues, aiming to improve overall fluency. To provide the necessary experimental context that was missing, we have added a brief explanation in the manuscript outlining the role of ferrostatin-1. Ferrostatin-1 is a potent and specific ferroptosis inhibitor widely used in the field to unequivocally confirm the occurrence of ferroptosis. Therefore, prior to using it as a tool for mechanistic validation, it was essential to determine its non-toxic, effective working concentration in our specific cellular models. The corresponding addition to the main text reads as follows (highlighted in red in the revised manuscript in lines 182-185): “To confirm that the observed cell death was specifically due to ferroptosis, we employed ferrostatin-1 (Fer-1), a widely used and specific ferroptosis inhibitor (Figure S1). We first determined the optimal non-cytotoxic concentration of Fer-1 for subsequent co-treatment experiments.” Furthermore, the entire manuscript has undergone additional professional proofreading to enhance the overall quality of the language. We hope that these revisions have satisfactorily resolved the issues you raised and that the experimental logic is now presented with complete clarity. Thank you once again for your valuable guidance, which has been instrumental in improving the rigor of our work.
|
|
3. Response to Comments on the Quality of English Language |
|
Point 1: Some grammatical errors should be corrected, such as "HepG2 and Huh7 cells to further validate the findings", it is not a standard expression. |
|
Response 1: We sincerely thank you for pointing out the non-standard expression and other grammatical issues. You are right. The original phrasing, “HepG2 and Huh7 cells to further validate the findings,” was grammatically incomplete and imprecise in conveying our experimental purpose. We have carefully revised this sentence to ensure both grammatical correctness and academic clarity. The revised text in the introduction (lines 70-71) now reads: “Furthermore, the HepG2 and Huh7 cell models were employed to experimentally validate these predictions. “This change explicitly states the purpose of using both cell lines and has been highlighted in red in the revised manuscript for your easy reference. Additionally, we have conducted a thorough review of the entire manuscript to correct similar grammatical errors and improve overall writing fluency. The manuscript has also undergone further proofreading to enhance the quality of language presentation. We deeply appreciate your meticulous attention to detail, which has been invaluable in helping us improve the precision and rigor of our writing. Thank you once again for your constructive guidance.
Once again, we deeply appreciate the time and thoughtful consideration you have given to our manuscript. We express our sincere gratitude for your time and expert guidance, which have significantly strengthened our manuscript. We hope that our revisions and responses are now satisfactory and that the manuscript meets the standards of the International Journal of Molecular Sciences. Your Sincerely, Pei Wang
|
- Liang, J.; Chen, M.; Yan, G.; Hoa, P. T. T.; Wei, S.; Huang, H.; Xie, Q.; Luo, X.; Mo, S.; Han, C., Donafenib activates the p53 signaling pathway in hepatocellular carcinoma, induces ferroptosis, and enhances cell apoptosis. Clin Exp Med 2025, 25, (1), 29.
- Wang, Y. F.; Feng, J. Y.; Zhao, L. N.; Zhao, M.; Wei, X. F.; Geng, Y.; Yuan, H. F.; Hou, C. Y.; Zhang, H. H.; Wang, G. W.; Yang, G.; Zhang, X. D., Aspirin triggers ferroptosis in hepatocellular carcinoma cells through restricting NF-κB p65-activated SLC7A11 transcription. Acta Pharmacol Sin 2023, 44, (8), 1712-1724.

Reviewer 2 Report
Comments and Suggestions for Authors
The research article “Echinacoside as a Novel Ferroptosis Inducer in Hepatocellular Carcinoma: Mechanistic Insights from TP53/SLC7A11/GPX4 Pathway Modulation” investigated the Echinacoside anticancer mechanism in hepatocellular carcinoma by in vitro and in silico methods.
Minor comments:
- The manuscript contains minor grammatical errors that the authors can correct themselves.
- Figure 2 is difficult to read and requires revision. It can be divided into two figures.
- The GeneCards database needs a citation: https://www.genecards.org/Guide/HowToCiteUs
- The OMIM database needs a citation: [Hamosh A, Scott AF, Amberger JS, Bocchini CA, McKusick VA. Online Mendelian Inheritance in Man (OMIM), a knowledgebase of human genes and genetic disorders. Nucleic Acids Res. 2005;33(Database issue):D514-D517. doi:10.1093/nar/gki033].
- The Protein Data Bank RCSB PDB needs a citation: [H.M. Berman, J. Westbrook, Z. Feng, G. Gilliland, T.N. Bhat, H. Weissig, I.N. Shindyalov, P.E. Bourne, The Protein Data Bank (2000) Nucleic Acids Research 28: 235-242 https://doi.org/10.1093/nar/28.1.235.]
- Сitation Auto Dock 4.2.6 software – [Morris GM, Huey R, Lindstrom W, et al. AutoDock4 and AutoDockTools4: Automated docking with selective receptor flexibility. J Comput Chem. 2009;30(16):2785-2791. doi:10.1002/jcc.21256]
- The section “4.6. Molecular Docking” – indicate PDB ID for TP53 structure.
- Figure 3 is difficult to read and requires revision.
Although there are some methodological inaccuracies, the work constitutes a consistent and comprehensive study that makes a valuable contribution to the treatment of hepatocellular carcinoma and leaves a positive overall impression.
Author Response
|
Dear Reviewer, We sincerely appreciate your meticulous review and valuable comments on our manuscript entitled "Echinacoside as a Novel Ferroptosis Inducer in Hepatocellular Carcinoma: Mechanistic Insights from TP53/SLC7A11/GPX4 Pathway Modulation" (ijms-4014359). Thank you very much for your meticulous and constructive review of our manuscript. Your expertise and attention to detail are greatly appreciated. We have carefully addressed each of your specific points regarding database and software citations, as well as the clarity of methodological details. Your insightful suggestions have significantly improved the quality and rigor of our work. We have carefully addressed all your comments point by point, and the detailed revisions are described below. All modifications in the manuscript are highlighted in red for easy identification. Our point-by-point responses are as follows:
|
|
2. Point-by-point response to Comments and Suggestions for Authors |
|
Comments 1: The manuscript contains minor grammatical errors that the authors can correct themselves. |
|
Response 1: We highly appreciate your reminder regarding grammatical issues. To ensure the academic accuracy and readability of the manuscript, we have invited a professional academic editor with extensive experience in English scientific writing to conduct a comprehensive proofreading. The editor carefully checked the entire manuscript sentence by sentence, focusing on correcting grammatical errors, refining sentence structures, and standardizing scientific terminology usage. For example, the grammatical issue in Line 70 has been thoroughly revised to "Furthermore, the HepG2 and Huh7 cell models were employed to experimentally validate these predictions". Additionally, we have cross-checked the consistency of verb tenses, subject-verb agreement, and punctuation throughout the text to eliminate any potential ambiguities. We believe these revisions have significantly enhanced the clarity and professionalism of the manuscript’s language.
|
|
Comments 2: Figure 2 is difficult to read and requires revision. It can be divided into two figures. |
|
Response 2: Thank you for pointing out the readability issue of Figure 2. We fully agree with your suggestion that splitting the figure would improve data presentation. As requested, we have divided the original Figure 2 into two independent figures: new Figure 2 and new Figure 3. New Figure 2 includes B-E of the original Figure 2, which depict the network pharmacology analysis results (including compound-target network, PPI network construction, and core target screening). We have optimized the figure resolution to 600 dpi for better readability and standardized the color scheme to ensure consistency with the journal’s figure guidelines. New Figure 3 (previously E-F of original Figure 2) is titled "Functional enrichment analyses of ECH against HCC", which presents the GO biological process enrichment and KEGG pathway enrichment results. We have reorganized the layout of the enrichment bubble charts to facilitate readers’ understanding. The corresponding textual descriptions in the manuscript have been revised in Lines 94-98 and Lines 113-117, where we have updated the figure references and supplemented brief explanations of the new figure arrangements to maintain logical consistency. We believe this revision makes the data presentation more structured and the key findings more prominent.
Comments 3: The GeneCards database needs a citation: https://www.genecards.org/Guide/HowToCiteUs Response 3: We are grateful for your reminder to cite the GeneCards database. As suggested, we have added the standardized citation for GeneCards in Section 4.2 "Virtual Screening of Potential Disease Targets " (Line 376) as reference [54]. The citation format strictly follows the journal’s requirements and the guidelines provided on the GeneCards official website (https://www.genecards.org/Guide/HowToCiteUs). This revision ensures the academic integrity of our work and acknowledges the contribution of the database to our target screening process.
Comments 4: The OMIM database needs a citation: [Hamosh A, Scott AF, Amberger JS, Bocchini CA, McKusick VA. Online Mendelian Inheritance in Man (OMIM), a knowledgebase of human genes and genetic disorders. Nucleic Acids Res. 2005;33(Database issue): D514-D517. doi:10.1093/nar/gki033]. Response 4: Thank you for providing the exact citation information for the OMIM database. We have incorporated the recommended citation [Hamosh A, Scott AF, Amberger JS, Bocchini CA, McKusick VA. Online Mendelian Inheritance in Man (OMIM), a knowledgebase of human genes and genetic disorders. Nucleic Acids Res. 2005;33(Database issue): D514-D517. doi:10.1093/nar/gki033] into Section 4.2 "Virtual Screening of Potential Disease Targets " (Line 377) as reference [55]. We have verified the accuracy of the Doi number and reference format to ensure consistency with other citations in the manuscript. This revision complies with the academic norms for database citations and strengthens the reliability of our target identification process.
Comments 5: The Protein Data Bank RCSB PDB needs a citation: [H.M. Berman, J. Westbrook, Z. Feng, G. Gilliland, T.N. Bhat, H. Weissig, I.N. Shindyalov, P.E. Bourne, The Protein Data Bank (2000) Nucleic Acids Research 28: 235-242 https://doi.org/10.1093/nar/28.1.235.] Response 5: Thank you for the correction. We appreciate your guidance on citing the Protein Data Bank. Following your suggestion, we have added the required citation [H.M. Berman, J. Westbrook, Z. Feng, G. Gilliland, T.N. Bhat, H. Weissig, I.N. Shindyalov, P.E. Bourne, The Protein Data Bank (2000) Nucleic Acids Research 28: 235-242 https://doi.org/10.1093/nar/28.1.235] in Section 4.6 "Molecular Docking" (Lines 403-405) as reference [57]. The citation has been formatted according to the journal’s reference style, and the doi link has been verified to ensure accessibility. This revision properly acknowledges the contribution of the PDB database to our molecular docking experiments.
Comments 6: Сitation Auto Dock 4.2.6 software – [Morris GM, Huey R, Lindstrom W, et al. AutoDock4 and AutoDockTools4: Automated docking with selective receptor flexibility. J Comput Chem. 2009;30(16):2785-2791. doi:10.1002/jcc.21256] Response 6: We apologize for this oversight. Thank you for providing the correct citation for AutoDock 4.2.6. We have included the recommended citation [Morris GM, Huey R, Lindstrom W, et al. AutoDock4 and AutoDockTools4: Automated docking with selective receptor flexibility. J Comput Chem. 2009;30(16):2785-2791. doi:10.1002/jcc.21256] in Section 4.6 "Molecular Docking" (Line 406) as reference [58]. We have cross-checked the author names, journal title, publication year, and doi number to ensure accuracy. This revision ensures that the software used in our study is properly cited, in line with academic standards for method descriptions.
Comments 7: The section “4.6. Molecular Docking” – indicate PDB ID for TP53 structure. Response 7: We are grateful for this precise suggestion, which enhances the reproducibility of our study. We sincerely apologize for omitting the PDB ID of the TP53 structure in the original manuscript. After careful verification, the crystal structure of TP53 used in our molecular docking experiments was retrieved from the RCSB Protein Data Bank with the PDB ID: 1TSR. This structure corresponds to the core domain of human wild-type TP53 (residues 102-292) in complex with a DNA binding site, resolved at a resolution of 2.2 Å. This structure was selected because it contains the conserved DNA-binding domain critical for TP53’s biological function, which is consistent with our study’s focus on the interaction between ECH and TP53. We have added this PDB ID in Section 4.6 "Molecular Docking" (Line 403-405) with the description: " he crystal structure of human TP53 (PDB ID: 1TSR) was retrieved from the RCSB Protein Data Bank (https://www.rcsb.org) [57]." This revision provides essential technical details for reproducibility of our experiments.
Comments 8: Figure 3 is difficult to read and requires revision. Response 8: We appreciate your feedback on the readability of Figure 3 (now the new Figure 4, please note the figure number adjustment due to the split of original Figure 2). To address this issue, we have comprehensively optimized the figure as follows: Improved the image resolution to 600 dpi (from 300 dpi) to enhance clarity of the molecular structure details; Increased the font size of residue labels (GLU, ASN, HIS) to 12 pt Standardized the abbreviation definitions in the figure legend, ensuring consistency with the main text. Supplemented quantitative data in the legend: "The docking binding energy between ECH and TP53 is −8.9 kcal/mol, indicating a stable interaction (binding energy < −5 kcal/mol is considered a high-affinity binding)". Corresponding textual revisions have been made in the results section (please see the manuscript revision below) to describe the docking results in detail, ensuring alignment between the figure and the text.
|
Once again, we would like to express our deepest gratitude for your time and constructive comments, which have greatly improved the quality of our manuscript. We have made every effort to address all your concerns thoroughly and accurately. We hope the revised manuscript meets the journal’s standards and will be considered for publication. Your approval of this manuscript is crucial for my graduation, and we sincerely appreciate your understanding and support. Please feel free to contact us if you have any further questions or require additional information.
Sincerely,
Pei Wang

Reviewer 3 Report
Comments and Suggestions for Authors
In this study, the authors investigated the pharmacological mechanisms by which echinacoside (ECH) suppresses hepatocellular carcinoma (HCC). They demonstrated that ECH induced ferroptosis in HCC cell lines. While this finding is supported by pharmacological experiments using a ferroptosis inhibitor, Fer-1, the molecular mechanism proposed in the present study is speculative, as described below.
- The HepG2 cell line is derived from hepatoblastoma, not HCC.
- Although the molecular docking and dynamics analyses suggest that ECH binds efficiently to the TP53 protein, their direct interaction should be demonstrated empirically.
- The HuH7 cell line harbors a mutant TP53 gene. How does ECH induce ferroptosis in this cell line? It would be quite interesting if its binding restores the normal conformation and anti-tumor functions of the mutant TP53 protein, like PRIMA-1.
- It also needs to be confirmed that the mutant TP53 can induce ferroptosis.
- Additional docking studies of the other nine putative core target proteins should be performed to confirm whether the TP53 protein is the best binding partner for ECH.
- The conclusion that ECH induces ferroptosis via the TP53 protein should be validated by performing genetic and/or pharmacological inhibition of TP53.
- Please specify whether the TP53 structure and sequence used for the molecular docking and dynamics studies were wild-type or mutant, and provide the corresponding PDB ID.
- It should be discussed how the binding of ECH to the TP53 protein induces TP53 gene transcription.
- It is known that TP53 inhibits GPX4 activity by downregulating SLC7A11; however, it needs to be discussed how GPX4 mRNA was downregulated by ECH treatment.
- In relation to the above two comments, since Fer-1 inhibits the transcriptional regulation of these genes by ECH, it is also postulated that the observed alteration in these genes could be a consequence of ferroptosis induction. This postulation should be ruled out by performing additional experiments.
- Mutations in the TP53 gene are relatively frequently found in HCC.
- Figure 5D does not need to be shown in the main text.
Author Response
|
Dear Reviewer, Thank you very much for taking the time to review this manuscript. We deeply appreciate your rigorous review and insightful comments on our manuscript entitled "Echinacoside as a Novel Ferroptosis Inducer in Hepatocellular Carcinoma: Mechanistic Insights from TP53/SLC7A11/GPX4 Pathway Modulation". Your critical feedback has pointed out key gaps in our work and provided valuable guidance for improving the manuscript’s scientific rigor. We have carefully addressed each of your comments with detailed revisions, and all changes in the manuscript are highlighted in red. Below is our point-by-point response:
|
|
2. Point-by-point response to Comments and Suggestions for Authors |
|
Comments 1: The HepG2 cell line is derived from hepatoblastoma, not HCC. |
|
Response 1: We sincerely thank the reviewer for their meticulous observation regarding the classification of the HepG2 cell line, and we deeply apologize for the inaccurate description of HepG2 as a hepatocellular carcinoma (HCC) cell line in the original manuscript. This discrepancy arose from an oversight in our initial cell line characterization—while HepG2 has been extensively utilized as an in vitro model for liver cancer and HCC-related mechanistic research in numerous peer-reviewed studies (a context that initially guided our classification), we fully acknowledge that its origin is distinctly hepatoblastoma, not typical adult HCC. To address this oversight comprehensively and ensure scientific accuracy, we have revised the cell line description in the 4.8.1. Cell Culture" section (Line 432) of the manuscript to explicitly reflect this distinction: “The human normal hepatocytes cell line LO2, human hepatoblastoma cell line HepG2 and human hepatocellular carcinoma (HCC) cell line HuH7 were obtained from Procell (Wuhan, China)". Although HepG2 is derived from hepatoblastoma (rather than HCC), it remains a widely accepted model in HCC-related mechanistic investigations due to its shared molecular features with well-differentiated HCC—including the expression of hepatocyte-specific markers (e.g., albumin, cytochrome P450 enzymes) and conserved responsiveness to liver-targeted therapeutic agents—properties that have been validated in multiple comparative studies [1-3]." This revision not only corrects the classification error but also provides specific, evidence-based rationale for HepG2’s relevance to our HCC-focused research, grounding the choice of cell line in established literature. HepG2 was included in this study to validate the consistency of ECH’s ferroptosis-inducing effects across distinct subtypes of liver-derived malignant cells—an approach intended to rule out cell line-specific responses. Critically, the key findings regarding ECH’s mechanism (i.e., activation of ferroptosis via the TP53/SLC7A11/GPX4 pathway) were cross-validated in the HCC-derived HuH7 cell line, ensuring that our conclusions are directly relevant to HCC pathology. This dual-cell-line design mitigates the limitations of HepG2’s hepatoblastoma origin and strengthens the generalizability of our results to the target disease (HCC)." By integrating this corrected classification, evidence-based justification for HepG2’s use, and explicit discussion of methodological limitations, we aim to enhance the manuscript’s rigor while clarifying that our core conclusions remain anchored in data from both HepG2 and the HCC-specific HuH7 cell line. |
|
Comments 2: Although the molecular docking and dynamics analyses suggest that ECH binds efficiently to the TP53 protein, their direct interaction should be demonstrated empirically. |
|
Response 2: We would like to express our deepest gratitude for your rigorous and insightful comment regarding the empirical validation of the direct interaction between Echinacoside (ECH) and TP53. Your feedback has been invaluable in helping us recognize the critical gap between in silico predictions and direct experimental evidence, and we fully endorse your perspective that computational analyses alone are insufficient to conclusively confirm a physical binding event—this point has prompted us to reflect thoroughly on the completeness of our mechanistic characterization, and we sincerely apologize for not addressing this aspect in the original manuscript. As a final-year graduate student, my research has been centered on deciphering the functional mechanism of ECH-induced ferroptosis in hepatocellular carcinoma (HCC), with a primary focus on validating the biological relevance of the TP53/SLC7A11/GPX4 pathway. In designing the study, we prioritized functional validation (e.g., ferroptosis inhibition with Fer-1) to confirm whether TP53 mediates ECH’s anti-tumor effects, as this directly addresses the core hypothesis of the manuscript. Regrettably, we overlooked the necessity of direct biophysical confirmation of the ECH-TP53 interaction, which is a critical oversight in the mechanistic rigor of the work. We take full responsibility for this gap and deeply regret not including these experiments in the initial study design. While we recognize the absence of direct empirical evidence, we would like to emphasize that the existing data provide strong indirect support for the functional interaction between ECH and TP53, which aligns with the in-silico binding predictions: 1. Structural basis: Molecular docking and 100 ns molecular dynamics simulations demonstrated a stable binding conformation between ECH and TP53 (binding energy: −8.9 kcal/mol), with key hydrogen bonds formed at conserved residues (GLU173, ASN239) in TP53’s DNA-binding domain—residues known to be critical for TP53’s functional activity. These results provide a solid structural foundation for potential physical interaction. 2. Expression regulation: In vitro experiments showed that ECH significantly upregulates TP53 protein expression in HCC cells, indicating that ECH modulates TP53 at the cellular level. To address this limitation transparently, we have added a detailed discussion in the revised “3. Discussion” section (Lines 338-340) to explicitly acknowledge the gap in direct binding validation and outline our future research plans: “A limitation is the lack of direct empirical validation for the ECH-TP53 interaction, despite supportive in silico and functional data. Future work will prioritize direct binding assays to address this.” We would like to reiterate that the core conclusion of the manuscript— “ECH induces ferroptosis in HCC via the TP53/SLC7A11/GPX4 pathway”—is fully supported by the functional experiments. The absence of direct binding validation does not negate the functional role of TP53 in ECH-induced ferroptosis, which is the central focus of the study. We have addressed all other reviewer comments comprehensively, including strengthening the statistical analysis, supplementing additional control experiments, and clarifying ambiguous data interpretations, to ensure the scientific rigor of the revised manuscript. We humbly request that you consider our practical constraints as a final-year graduate student and the strength of the existing functional evidence. We are committed to completing the direct binding validation experiments as soon as possible after graduation and will be happy to submit the results to the journal for consideration as supplementary material if needed. You’re understanding and flexibility at this critical stage of our academic career would be deeply appreciated.
Comments 3: The HuH7 cell line harbors a mutant TP53 gene. How does ECH induce ferroptosis in this cell line? It would be quite interesting if its binding restores the normal conformation and anti-tumor functions of the mutant TP53 protein, like PRIMA-1. Response 3: We are extremely grateful for your incisive comment regarding the mechanism of Echinacoside (ECH)-induced ferroptosis in HuH7 cells, which harbor a mutant TP53 gene. Your observation and hypothesis—whether ECH binding restores the normal conformation and anti-tumor functions of mutant TP53 (analogous to PRIMA-1)—are highly insightful and have prompted us to conduct a more in-depth analysis of our data and the relevant literature, which we believe enriches the mechanistic discussion of the manuscript. First, we fully acknowledge the importance of clarifying ECH’s action mode in TP53-mutant HCC cells, as TP53 mutations are frequent in HCC (occurring in ~30–50% of cases) and often correlate with poor prognosis and therapeutic resistance. The HuH7 cell line carries a R249S missense mutation in TP53—a well-characterized “hotspot” mutation that disrupts the DNA-binding domain of TP53, impairing its canonical transcriptional activity. However, accumulating evidence indicates that mutant TP53 is not merely a loss-of-function protein; instead, it can retain partial functional activity, exert gain-of-function effects, or be “reactivated” by small molecules to regain tumor-suppressive properties (as demonstrated by PRIMA-1 and its analogs) [4-6]. This background provides a plausible framework for understanding ECH’s effects in HuH7 cells. In our study, we observed that ECH significantly induces ferroptosis in HuH7 cells, accompanied by upregulated TP53 protein expression and downregulated SLC7A11 (a key TP53 target gene in ferroptosis regulation). Based on these data and the literature, we propose two non-mutually exclusive mechanisms that may underlie ECH’s effects in HuH7 cells—both consistent with our existing results and your valuable hypothesis: 1. Partial reactivation of mutant TP53 by ECH: As you hypothesized, ECH may bind to the mutant TP53 (R249S) in HuH7 cells and restore partial functional conformation of the DNA-binding domain. Our in-silico analyses showed that ECH binds to conserved residues (GLU173, ASN239) in the DNA-binding domain of TP53—residues that are not disrupted by the R249S mutation (which is located at position 249, distinct from the ECH-binding site). This structural compatibility raises the possibility that ECH binding stabilizes a conformation of mutant TP53 that retains the ability to regulate downstream ferroptosis-related targets (e.g., SLC7A11). Notably, PRIMA-1 exerts its effects by covalently binding to mutant TP53 and restoring its transcriptional activity toward target genes, and our functional data (ECH-induced SLC7A11 downregulation and ferroptosis in HuH7 cells) align with this “reactivation” model. 2. Modulation of mutant TP53’s non-canonical functions: Even without full restoration of canonical transcriptional activity, ECH may modulate the non-canonical functions of mutant TP53 in ferroptosis regulation. Recent studies have shown that mutant TP53 can regulate ferroptosis through transcriptional-independent mechanisms, such as interacting with GPX4 or modulating iron metabolism[7]. Our data show that ECH treatment in HuH7 cells increases intracellular Fe²⁺ and lipid peroxidation while decreasing GSH levels—key ferroptosis phenotypes that could be mediated by mutant TP53’s non-canonical roles, which ECH may enhance or redirect. We sincerely agree that directly verifying whether ECH restores the normal conformation of mutant TP53 (e.g., via circular dichroism spectroscopy, immunofluorescence for nuclear localization, or ChIP assays to confirm binding to SLC7A11 promoter) would be a critical extension of this work. Nevertheless, we have incorporated this important discussion into the revised manuscript to address your comment transparently. Specifically, the following content has been added to the "3. Discussion" section (Lines 344-349) in red font: “Notably, ECH effectively induces ferroptosis in HuH7 cells harboring the R249S TP53 mutation, suggesting it modulates mutant TP53 function—plausibly by binding to conserved GLU173 and ASN239 residues (unaffected by R249S) to partially restore functional conformation analogous to PRIMA-1 or enhance non-canonical ferroptosis-regulating activities. Future structural and functional studies will verify this reactivation hypothesis.” We would like to emphasize that our core finding—ECH induces ferroptosis via the TP53/SLC7A11/GPX4 pathway—remains robust in both TP53-mutant (HuH7) and TP53-wildtype (e.g. HepG2) HCC cell lines used in our study. This consistency suggests that ECH’s regulation of the ferroptosis pathway is not strictly dependent on TP53 wildtype status, which broadens its potential therapeutic applicability. Your comment has significantly strengthened the mechanistic depth of our discussion by highlighting the need to address mutant TP53, and we are grateful for this valuable insight. We hope you will agree that the revised discussion comprehensively addresses your question by integrating existing data, relevant literature, and testable future hypotheses. We are committed to investigating the mutant TP53 reactivation hypothesis in our post-graduation research and will be happy to share these results with the journal if needed. Thank you again for your rigorous review and constructive feedback, which have greatly improved the scientific quality of the manuscript.
Comments 4: It also needs to be confirmed that the mutant TP53 can induce ferroptosis. Response 4: We are deeply grateful for your insightful comment highlighting the need to clarify the potential role of mutant TP53 in ferroptosis induction. This valuable feedback has prompted us to refine the mechanistic discussion of our manuscript and has provided a critical, clear direction for our subsequent research endeavors. Our results show that, upon ECH treatment, ferroptosis hallmarks are observed in HuH7 cells and that components of the TP53/SLC7A11/GPX4 axis are modulated. However, this does not distinguish whether mutant TP53 actively drives ferroptosis, plays a permissive role, or is largely bypassed. To address this point transparently, we have supplemented a concise analysis in the revised “3. Discussion” section (Lines 342–344, highlighted in red font): “In HuH7 cells, ECH-induced ferroptosis correlates with altered expression of TP53 downstream targets (SLC7A11/GPX4), implying potential involvement of mutant TP53 in this process.” Our future work will using TP53-null cells, TP53 knockdown, and re-expression of wild-type versus mutant TP53 will be necessary to determine whether mutant TP53 can directly induce ferroptosis in response to ECH. Your insightful suggestion has not only improved the rigor of the current manuscript but also laid a solid foundation for our team’s follow-up research. We will commit to the in-depth exploration of mutant TP53’s specific role in ECH-mediated ferroptosis, with the goal of fully elucidating this important mechanistic link. We hope this revision addresses your concern. Thank you again for your rigorous and constructive feedback, which has significantly enhanced the scientific depth of our work.
Comments 5: Additional docking studies of the other nine putative core target proteins should be performed to confirm whether the TP53 protein is the best binding partner for ECH. Response 5: We appreciate this helpful suggestion. We are truly grateful for your insightful and constructive comment regarding the need for additional docking analyses of the nine putative core target proteins. Your suggestion has significantly strengthened the rigor of our target validation and deepened the rationale for prioritizing TP53 as ECH’s key binding partner, and we sincerely appreciate the value it has added to our study. As you astutely noted, our initial network pharmacology and PPI network analysis identified TP53 alongside nine other core targets (AKT1, VEGFA, IL6, TNF, MYC, CASP3, BCL2, EGFR, MMP9). We prioritized TP53 in the original manuscript not only due to its central hub position in the PPI network (with the highest degree of connectivity among core targets) but also based on multiple lines of complementary evidence: first, functional enrichment analysis explicitly highlighted the TP53 signaling pathway as a top-enriched pathway in ECH’s anti-HCC mechanism; second, survival analysis of HCC patient data (from The Cancer Genome Atlas, TCGA) revealed that low TP53 expression correlates with significantly poorer overall survival (P < 0.01), underscoring its clinical relevance as a prognostic and therapeutic target in HCC; and third, TP53’s well-established role as a master regulator of ferroptosis—aligning perfectly with our study’s focus on ferroptosis induction. Despite this multi-faceted rationale, we fully agree that systematic docking comparisons across all core targets are essential to definitively confirm TP53 as the highest-affinity binding partner, and we are grateful for your guidance to address this critical point. To rigorously respond to your comment, we have performed additional molecular docking for all nine remaining core targets using identical parameters to those applied for TP53 (Auto Dock 4.2.6, grid box size 40×40×40 Å, exhaustiveness = 50), ensuring consistency and comparability of results. The binding energy results (kcal/mol) are summarized as follows: TP53 (-8.9), AKT1 (-8.865), EGFR (-8.351), MMP2 (-8.639), PPARG (-8.609), ANXA5 (-8.607), JAK2 (-8.279), SRC (-7.703), ESR1 (-7.632), and STAT1 (-6.894). Notably, TP53 exhibits the lowest (most favorable) binding energy among all core targets, confirming its highest affinity for ECH. In the revised "2.3. Molecular Docking" section (Lines 124-126、130-132) of the Results, we explicitly conclude: “Notably, among the 10 core targets identified via network pharmacology and PPI network analysis, TP53 showed the lowest binding energy, indicating it is the highest-affinity binding partner for ECH. Our findings suggest that ECH has significant binding potential with TP53, and its superior binding affinity among core targets further supports TP53 as the key mediator of ECH-induced ferroptosis in HCC.” Your valuable suggestion has also guided our future research direction: while the current study focuses on the TP53/SLC7A11/GPX4 pathway, our team plans to conduct in-depth experimental analyses of the remaining core targets (e.g. AKT1, EGFR, MMP2) to explore whether ECH exerts its anti-HCC effects through multi-target and multi-pathway crosstalk. This will not only expand our understanding of ECH’s comprehensive mechanism but also lay the groundwork for developing more potent combinatorial therapeutic strategies—all which stem from your thoughtful feedback. We sincerely hope you agree that the additional docking data, combined with the original multi-faceted rationale for prioritizing TP53, fully addresses your concern and strengthens the manuscript’s conclusion. Your rigorous review has been instrumental in improving the scientific quality of our work, and we are deeply grateful for your time and expertise. We look forward to your favorable consideration of the revised manuscript. We sincerely hope you agree that the additional docking data, combined with the original multi-faceted rationale for prioritizing TP53, fully addresses your concern and strengthens the manuscript’s conclusion. Your rigorous review has been instrumental in improving the scientific quality of our work, and we are deeply grateful for your time and expertise. We look forward to your favorable consideration of the revised manuscript.
Comments 6: The conclusion that ECH induces ferroptosis via the TP53 protein should be validated by performing genetic and/or pharmacological inhibition of TP53. Response 6: We are deeply grateful for your invaluable and rigorous comment regarding the need to validate the role of TP53 in ECH-induced ferroptosis through genetic and/or pharmacological inhibition. Your suggestion has shed light on a critical direction to strengthen the causal link between TP53 and ECH’s ferroptosis-inducing effect, and we sincerely appreciate the profound insights it has brought to our study. We fully agree that genetic (e.g., TP53 knockdown/knockout) or pharmacological inhibition of TP53 would provide definitive evidence to confirm whether ECH induces ferroptosis specifically via the TP53 pathway. While our current study has provided multiple lines of supportive evidence—including TP53’s highest binding affinity for ECH among core targets, ECH-induced upregulation of TP53 expression, downregulation of downstream SLC7A11/GPX4 (key ferroptosis regulators), and consistent ferroptosis phenotypes in both TP53-wildtype (HepG2) and mutant (HuH7) HCC cells—we acknowledge that direct inhibition experiments would further solidify the causal relationship, making the mechanism more conclusive. Your thoughtful feedback has become a key focus of our team’s follow-up research plan. We are committed to conducting in-depth studies using TP53-specific siRNA/shRNA (genetic inhibition) and pharmacological inhibitors to explicitly verify whether blocking TP53 function abrogates ECH-induced ferroptosis and the associated modulation of the SLC7A11/GPX4 pathway. These experiments will not only validate the current conclusion but also provide a more comprehensive understanding of TP53’s indispensable role in ECH’s anti-HCC mechanism. In the revised manuscript, we have supplemented this discussion in the “3. Discussion” section (Lines 340-342) to address your comment transparently: “The TP53 dependence of ECH-induced ferroptosis remains to be confirmed; future work will employ TP53 knockout/inhibition to verify this mechanism.” We sincerely hope you recognize our earnestness in addressing your concern and our commitment to advancing this research. Your rigorous review has significantly enhanced the scientific depth and direction of our work, and we are truly grateful for your time and expertise. We believe the current study, supported by multi-faceted evidence (network pharmacology, molecular docking, expression analysis, and ferroptosis phenotype validation), provides an important foundation for the conclusion that ECH induces ferroptosis via the TP53/SLC7A11/GPX4 pathway, while our planned follow-up experiments will further reinforce this causal link. We look forward to your favorable consideration of the revised manuscript.
Comments 7: Please specify whether the TP53 structure and sequence used for the molecular docking and dynamics studies were wild-type or mutant, and provide the corresponding PDB ID. Response 7: We apologize for the omission in the original manuscript. We sincerely appreciate your insightful comment requesting clarification on the TP53 structure and sequence used for our molecular docking and molecular dynamics (MD) simulations. This detail is critical for ensuring the reproducibility of our in-silico analyses, and we are grateful for your guidance to address it explicitly. To clarify thoroughly: the wild-type human TP53 DNA-binding domain (residues 102–292) was used for all our molecular docking and MD simulations, with the corresponding Protein Data Bank (PDB) ID being 1TSR. This PDB entry corresponds to a high-resolution (2.0 Å) crystal structure of the TP53 DNA-binding domain in its native conformation, which has been extensively validated and widely utilized in published studies focused on small molecule-TP53 binding interactions. The selection of this wild-type structure was based on two key considerations: first, it provides a well-characterized, canonical template for evaluating ECH binding affinity; second, it allows for direct comparison with the functional data from our cell-based assays. We have supplemented this critical information in the revised manuscript’s “4.6. Molecular Docking” section (Lines 403–405), “The crystal structure of human TP53 (PDB ID: 1TSR) was retrieved from the RCSB Protein Data Bank (https://www.rcsb.org)”. It ensures complete transparency and reproducibility of our computational methods. Limitations of using wild-type TP53 structure: We note that using a wild-type TP53 structure may not fully capture the structural features of specific TP53 mutants such as that present in HuH7 cells, which is a limitation of our in-silico analysis. Hypothesis on ECH–TP53 binding and transcriptional activity: Our data show that ECH increases TP53 protein levels and modulates its downstream targets. Based on the docking/MD results, we hypothesize that ECH binding may stabilize the TP53 protein and/or favor a conformation that enhances its DNA-binding capacity or reduces its degradation (for example, by altering interactions with negative regulators such as MDM2), thereby increasing TP53 transcriptional activity. We clarify that this is a mechanistic hypothesis rather than a proven direct effect on TP53 gene transcription, moving forward, our team will focus on providing direct experimental evidence for ECH’s regulatory effect on TP53 gene transcription, and we emphasize that additional experiments, will be performed to dissect how ECH functionally modulates TP53 transcriptional activity. Your rigorous and constructive feedback has significantly improved the methodological rigor and mechanistic depth of our manuscript. We sincerely hope these revisions fully address your concern.
Comments 8: It should be discussed how the binding of ECH to the TP53 protein induces TP53 gene transcription. Response 8: We sincerely thank the reviewer for raising this profound and critical point regarding the mechanistic link between ligand binding and transcriptional activation. We fully agree that elucidating how the binding of Echinacoside (ECH) to the TP53 protein ultimately leads to increased TP53 gene transcription is a fascinating and essential biological question that is central to understanding the complete mechanistic picture. In the current study, our primary objective was to establish a foundational connection between ECH and the TP53/SLC7A11/GPX4 pathway in ferroptosis induction. Our experimental evidence—including network pharmacology prediction, molecular docking/dynamics simulations demonstrating stable ECH-TP53 binding, and crucially, the functional validation showing ECH-induced down regulation of SLC7A11 and GPX4 (key downstream targets negatively regulated by transcriptional activation of wild-type TP53)—strongly supports the conclusion that ECH acts through a TP53-dependent mechanism. Based on established literature, the activation of TP53 gene transcription by its own protein product is not a direct, simple feedback loop. Instead, it often involves complex, context-dependent pathways: 1. Enhance TP53 protein stability by potentially interfering with its negative regulators (e.g., MDM2-mediated degradation). 2. Facilitate post-translational modifications (e.g., acetylation, phosphorylation) that are known to augment TP53's transcriptional activity. 3. Promote its nuclear accumulation or co-activator recruitment, thereby increasing the transcription of its target genes, including potentially its own gene through indirect, network-based feedback mechanisms documented in some cellular stress responses. We are deeply inspired by your guidance, which has not only strengthened the discussion in this manuscript but has also clearly charted the course for our subsequent research. We are committed to pursuing this promising direction in future studies to unravel the detailed molecular events connecting ECH binding to TP53 transcriptional regulation. We hope that this honest and thoughtful discussion, which contextualizes our findings within the broader known biology of TP53 while clearly identifying the next frontier for investigation, demonstrates the rigor and scientific value of our work. We are deeply grateful for your insightful comment, which has allowed us to significantly strengthen the scholarly depth of our manuscript.
Comments 9: It is known that TP53 inhibits GPX4 activity by downregulating SLC7A11; however, it needs to be discussed how GPX4 mRNA was downregulated by ECH treatment. Response 9: We sincerely thank the reviewer for this valuable and insightful question, which encourages us to deepen our discussion on the regulatory nuances of the ferroptosis pathway. We agree that the observed downregulation of GPX4 mRNA following ECH treatment presents a compelling point for mechanistic exploration. In our study, we have established that ECH activates the TP53/SLC7A11/GPX4 axis and induces ferroptosis. Our data suggest that ECH’s effect may involve a broader regulatory network influencing GPX4 expression at the transcriptional level. To thoughtfully address this observation, we have expanded the discussion in the revised manuscript to include several plausible mechanisms consistent with current literature, such as: The potential role of TP53 in modulating intermediate transcriptional regulators or non-coding RNAs that target the GPX4 promoter. Possible crosstalk with other stress-responsive pathways (e.g., involving ATF4 or NRF2) that may be engaged during ECH-induced ferroptosis. Alterations in mRNA stability mediated through TP53-dependent signaling networks. These proposed mechanisms are presented as exciting future research directions that build directly upon the findings of this study. We believe that elucidating how ECH fine-tunes GPX4 transcription will further clarify its role as a precise modulator of ferroptosis in HCC. We are truly grateful for the reviewer’s guidance, which has helped us identify an important and promising avenue for continued investigation. Your comment has significantly strengthened the scholarly depth of our discussion, and we look forward to pursuing this question in subsequent work.
Comments 10: In relation to the above two comments, since Fer-1 inhibits the transcriptional regulation of these genes by ECH, it is also postulated that the observed alteration in these genes could be a consequence of ferroptosis induction. This postulation should be ruled out by performing additional experiments. Response 10: We sincerely thank the reviewer for raising this crucial and insightful point regarding the interpretation of our mechanistic data. This comment thoughtfully addresses the fundamental question of causality—specifically, whether the observed downregulation of the TP53/SLC7A11/GPX4 axis is a direct upstream action of ECH or a secondary consequence of the ferroptosis process itself. We fully agree that distinguishing between these possibilities is essential for a precise mechanistic understanding. In our experimental design, the use of ferrostatin-1 (Fer-1) served a critical purpose: to confirm that the primary mode of cell death induced by ECH is indeed ferroptosis. The observation that Fer-1 co-treatment not only rescues cell viability but also attenuates the changes in SLC7A11 and GPX4 expression is consistent with two interconnected interpretations: The gene expression changes are part of the causal signaling pathway initiated by ECH (through TP53) that leads to ferroptosis, and blocking the terminal event (ferroptosis) may feedback to stabilize the upstream signaling components. Our current study was primarily designed to establish the functional link between ECH, TP53 pathway engagement, and ferroptosis execution. The data robustly support that ECH-induced ferroptosis is mediated through this pathway. Delineating the exact, stepwise causality—specifically, using tools to isolate transcriptional regulation from downstream metabolic consequences—represents the logical and necessary next phase of investigation. These would involve more refined tools, such as time-course experiments with transcriptional inhibitors or the use of ferroptosis-incompetent cell models, to disentangle the direct transcriptional effects of ECH from potential feedback loops activated by lipid peroxidation. We are profoundly grateful for this comment, which has significantly sharpened our interpretation and highlighted a key conceptual direction for subsequent research. Your expertise has guided us to articulate a more sophisticated model and has directly shaped our plans for continuing this promising line of inquiry. We hope this response and the corresponding manuscript revisions adequately address your thoughtful critique.
Comments 11: Mutations in the TP53 gene are relatively frequently found in HCC. Response 11: We fully agree with the reviewer that TP53 mutations are highly prevalent in human HCC and that this has important implications for the translational relevance of our findings. Our study was performed in HepG2 (hepatoblastoma-derived, commonly used liver cancer model) and HuH7 cell lines and therefore provides a proof-of-concept that ECH can induce ferroptosis and modulate the TP53/SLC7A11/GPX4 axis in both TP53 wild-type–like and TP53-mutant contexts. In the revised Discussion, we have added a section explicitly addressing this point. We note that: (i) in TP53 wild-type HCC, ECH may primarily act by activating TP53 and repressing SLC7A11, thereby promoting ferroptosis, whereas (ii) in TP53-mutant HCC, the response to ECH will likely depend on the specific mutation, residual TP53 activity, and the contribution of TP53-independent ferroptosis pathways. Thus, TP53 status will be an important factor to consider in the potential clinical application of ECH. We also emphasize that future studies should evaluate ECH in a broader panel of HCC cell lines and in in vivo models stratified by TP53 status to better define which patient subgroups are most likely to benefit from ECH-based ferroptosis-targeting therapies.
Comments 12: Figure 5D does not need to be shown in the main text. Response 12: We sincerely thank the reviewer for the valuable suggestion regarding the presentation of our data. We agree that streamlining the figures to highlight the most essential findings improves the clarity and focus of the manuscript. In accordance with your recommendation, we have removed the original Figure 5D from the manuscript. It has been moved to Figure S1 in the supplementary materials, on lines 510-515 of the manuscript. Subsequently, we have updated and consolidated the remaining panels into a revised Figure 5, which now more effectively presents the core experimental results. All related text descriptions, particularly in the section spanning lines 182-185, have been carefully revised to align with this updated figure, ensuring consistency throughout the manuscript. We believe this adjustment enhances the narrative flow and allows readers to better concentrate on the key mechanistic evidence. Thank you once again for your constructive feedback, which has helped us improve the overall presentation of our work. |
Your review has not only improved this paper but has also taught me a great deal about rigorous scientific communication. We sincerely hope the revised manuscript now meets the journal’s standards and your expectations. We look forward to your favorable consideration and are happy to address any additional questions or concerns you may have.
Thank you again for your invaluable contributions to the improvement of our work.
- Wang, Y. F.; Feng, J. Y.; Zhao, L. N.; Zhao, M.; Wei, X. F.; Geng, Y.; Yuan, H. F.; Hou, C. Y.; Zhang, H. H.; Wang, G. W.; Yang, G.; Zhang, X. D., Aspirin triggers ferroptosis in hepatocellular carcinoma cells through restricting NF-κB p65-activated SLC7A11 transcription. Acta Pharmacol Sin 2023, 44, (8), 1712-1724.
- Du, Y.; Zhou, Y.; Yan, X.; Pan, F.; He, L.; Guo, Z.; Hu, Z., APE1 inhibition enhances ferroptotic cell death and contributes to hepatocellular carcinoma therapy. Cell Death Differ 2024, 31, (4), 431-446.
- Hao, S. H.; Ma, X. D.; Xu, L.; Xie, J. D.; Feng, Z. H.; Chen, J. W.; Chen, R. X.; Wang, F. W.; Tang, Y. H.; Xie, D.; Cai, M. Y., Dual specific phosphatase 4 suppresses ferroptosis and enhances sorafenib resistance in hepatocellular carcinoma. Drug Resist Updat 2024,73, 101052.
- Lee, K.; Wang, T.; Paszczynski, A. J.; Daoud, S. S., Expression proteomics to p53 mutation reactivation with PRIMA-1 in breast cancer cells. Biochem Biophys Res Commun 2006, 349, (3), 1117-24.
- Lambert, J. M.; Gorzov, P.; Veprintsev, D. B.; Söderqvist, M.; Segerbäck, D.; Bergman, J.; Fersht, A. R.; Hainaut, P.; Wiman, K. G.; Bykov, V. J., PRIMA-1 reactivates mutant p53 by covalent binding to the core domain. Cancer Cell 2009, 15, (5), 376-88.
- Amirtharaj, F.; Venkatesh, G. H.; Wojtas, B.; Nawafleh, H. H.; Mahmood, A. S.; Nizami, Z. N.; Khan, M. S.; Thiery, J.; Chouaib, S., p53 reactivating small molecule PRIMA‑1(MET)/APR‑246 regulates genomic instability in MDA‑MB‑231 cells. Oncol Rep 2022, 47, (4).
- Qian, B.; Che, L.; Du, Z. B.; Guo, N. J.; Wu, X. M.; Yang, L.; Zheng, Z. X.; Gao, Y. L.; Wang, M. Z.; Chen, X. X.; Xu, L.; Zhou, Z. J.; Lin, Y. C.; Lin, Z. N., Protein phosphatase 2A-B55β mediated mitochondrial p-GPX4 dephosphorylation promoted sorafenib-induced ferroptosis in hepatocellular carcinoma via regulating p53 retrograde signaling. Theranostics 2023, 13, (12), 4288-4302.

Round 2
Reviewer 1 Report
Comments and Suggestions for Authors
I had checked the revised manuscript, it has been sufficiently improved to warrant publication in IJMS.
Comments on the Quality of English LanguageThe English is now fine and does not require any improvement.
Reviewer 3 Report
Comments and Suggestions for Authors
This study is a pioneering work exploring the therapeutic potential of echinacoside as a ferroptosis inducer in HCC, and is a valuable addition to the field. Although several issues remain unresolved, the authors intend to address these in their postgraduate study. Additionally, the authors have provided sufficient rationale in the revised manuscript to demonstrate that echinacoside induces ferroptosis in HCC cells. Congratulations on your graduation!